# Facile Synthesis of PVP-Coated Silver Nanoparticles and Evaluation of Their Physicochemical, Antimicrobial and Toxic Activity

Francisco N. Souza Neto [1,2,*], Leonardo A. Morais [1], Luiz F. Gorup [2,3,4,5,*], Lucas S. Ribeiro [2], Tassia J. Martins [2], Thayse Y. Hosida [1], Patricia Francatto [2], Debora B. Barbosa [1], Emerson R. Camargo [2] and Alberto C. B. Delbem [1,*]

[1] Araçatuba Dental School, Department of Pediatric Dentistry and Public Health, São Paulo State University (UNESP), Rua José Bonifácio 1193, Araçatuba 16015-050, SP, Brazil

[2] Department of Chemistry, Interdisciplinary Laboratory of Electrochemistry and Ceramics (LIEC), Federal University of São Carlos (UFSCar), Washington Luis Highway, km 235, São Carlos 13565-905, SP, Brazil; tassiajoi@gmail.com (T.J.M.)

[3] School of Chemistry and Food Science, Federal University of Rio Grande (FURG), Av. Italia km 8, Rio Grande 96203-900, RS, Brazil

[4] Materials Engineering, Campus Porto, Federal University of Pelotas (UFPel), Pelotas 96010-610, RS, Brazil

[5] Institute of Chemistry, Federal University of Alfenas (UNIFAL), Alfenas 37130-001, MG, Brazil

* Correspondence: francisco_nsn@yahoo.com.br (F.N.S.N.); lfgorup@gmail.com (L.F.G.); alberto.delbem@unesp.br (A.C.B.D.)

**Abstract:** This study focuses on the synthesis of silver nanoparticles (AgNPs) at different high concentrations and investigates their physicochemical properties, antimicrobial activity, and cytotoxicity. AgNPs were synthesized using the alcohol reduction process, involving the reduction of $AgNO_3$ and its subsequent stabilization via PVP at 80 °C for 4 h. The $AgNO_3$/PVP molar ratio and the average molecular weight were modified in this study. Characterization analyses revealed that the synthesized AgNPs exhibited characteristic surface plasmon resonance absorption peaks at approximately 415 nm, as observed in the UV–Vis spectrum. The results presented in X-ray diffractograms confirmed the face-centered cubic structure of metallic Ag in the nanoparticles. The nanoparticles demonstrated uniform size and shape, with controllable dimensions ranging from 3 to 800 nm. Regarding antimicrobial activity, the MIC solutions exhibited higher potency against the planktonic cells of *Candida albicans*. The determination of inhibition halos indicated that the silver nanoparticles had an impact on the microorganisms *Streptococcus mutans*, *Candida albicans*, and *Actinomyces israelii*. Furthermore, lower-concentration compositions showed reduced cytotoxic effects compared to higher-concentration particles. Based on the findings, it was concluded that the $AgNO_3$/PVP molar ratio plays a crucial role in the production of AgNPs. These synthesized nanoparticles exhibit desirable physicochemical properties and demonstrate potential antimicrobial activity and controlled cytotoxicity.

**Keywords:** silver nanoparticles; alcohol reduction process; Ostwald ripening; size control; antimicrobial properties

## 1. Introduction

In recent years, metallic nanoparticles have been widely investigated with respect to their ability to serve as replacements for antibiotics due to the increasing number of microorganisms acquiring resistance to these drugs [1–3]. The use of silver nanoparticles (AgNPs) is an alternative approach for treating microbial infections caused by antibiotic-resistant strains [4–6]. In particular, AgNPs have been used in several medical applications such as sunscreens, burn treatment, wound dressings, textiles, dental materials, bone implants, and medical device coatings [7–11].

AgNPs exhibit a broad range of antimicrobial activity against various microorganisms [12–15], and their antimicrobial activities, which are characteristic of silver ions ($Ag^+$), present a great affinity for nucleophilic atoms such as sulfur and phosphorus, which form structures in the cell membrane and inside cells [16]. Hence, the interaction between AgNPs and these biological structures can deregulate the cellular respiration process and interrupt the activity of adenosine triphosphate (ATP) [17] molecules and the cause of damage to deoxyribonucleic acid (DNA) replication in cell division, leading to the death of a microorganism [18,19].

The properties are very sensitive to the size, size distribution, and shapes of silver nanoparticles, so it is crucial to prepare silver nanoparticles of controllable, monodisperse sizes [20–23]. Currently, it is possible to synthesize AgNPs with good control of their sizes and structures. However, there are few chemical [24], physical [25], or biological [26] methods capable of producing uniformly sized and highly concentrated AgNP dispersions.

Surfactants and polymers are employed to control the shapes and sizes of nanoparticles [27]. The stabilization of AgNPs using a polymer is known as the steric method [28], in which a solution is achieved by binding polymeric molecules containing long alkyl chains to the particle surface [29]. This mechanism generally produces spherical particles due to their low surface energy [30].

The most common synthetic procedure for obtaining AgNPs was described by Turkevich et al. (1951) [31]. Dubbed the Turkevich Method, it is an effective method for the synthesis of nanoparticles using sodium citrate as a reducing agent for $Ag^+$ ions [32]. The disadvantage of Turkevich method is that it only permits the acquisition of particles at low concentrations and with a broad size distribution [33]. New methodologies are being described to solve this problem, and the reduction of silver ions using alcohol has attracted significant scientific interest [34–36]. Liz-Marzán et al., 1996 [37], Hah and Koo (2003) [38], and Almatroudi (2020) [39] showed that $Ag^+$ ions can be reduced via ethanol in the presence of different surfactants, resulting in particles with a narrow size distribution, reduced toxicity, and improved biomedical activities [40].

Some advantages have been reported regarding the synthetic method for the development of AgNPs in an alcohol reduction process. Javed et al. 2020 [41] mentioned that the use of poly(vinylpyrrolidone) (PVP) as a stabilizing agent and ethyl alcohol as a reducing agent can improve the monodispersing of the nanoparticles and enhance the reduction process of $Ag^+$ to $Ag^0$. In addition, PVP plays an important role in controlling the sizes of nanoparticles since this polymer acts as both a protective and coordinative agent [42]. The synthetic procedure employing the alcohol reduction process can be classified as a green chemistry strategy when the solvent used is also a reducing agent, as this decreases the number of reactants.

Currently, it is possible to produce AgNPs with good control of their sizes and structures. However, there are very few methods capable of producing high-concentration non-aqueous dispersions of uniform AgNPs with sizes below 10 nm. Since most alcohol-based wet chemical methods are heterogeneous nucleation processes, it is difficult to prepare AgNPs at high concentrations with monodisperse small sizes in one reaction system [43].

While the synthesis of AgNPS is a well-established topic in academic literature, the alcohol reduction process continues to hold promise for new discoveries. When combined with its cost-effectiveness and utilization of an eco-friendly solvent (i.e., aligning with the principles of green chemistry). Thus, the objective of this study was to obtain stable AgNPs in high concentrations with a controlled size, a narrow size distribution, and controllable structural and morphological properties as well as antimicrobial activity and cytotoxicity.

## 2. Materials and Methods

### 2.1. Synthesis of Silver Nanoparticles (AgNPs)

AgNPs were prepared by the alcohol reduction process. This method consists of the reduction of the silver precursor in an alcoholic medium and its subsequent stabilization by

polymer as described by Lee and Oh (2015) [44]. The detailed methodology can be found in the Supplementary Materials.

To evaluate the influence of the concentration and the average molecular weight (MW) on the AgNP synthesis process, triplicate experiments were performed regarding the initial reaction conditions (AgNO$_3$ and PVP 90 mmol L$^{-1}$), in which molar proportions of 1:4, 1:2, 2:1 and 4:1 AgNO$_3$:PVP (Ag:PVP) were prepared using PVP with MW = 10.000, 40.000 and 360.000 (10 K, 40 K e 360 K).

### 2.2. Physicochemical Characterization

The optical, structural, morphological, and microbiological properties of AgNPs were characterized by the following techniques: UV–Vis spectroscopy (UV–Vis), X-ray diffraction (XRD), dynamic light scattering (DLS), zeta potential, high-resolution transmission electron microscopy (HRTEM), Scanning Electron Microscopy (SEM), inhibition halo, minimum inhibitory concentration (MIC) and cellular viability.

XRD: AgNPs were characterized in the 2θ range from 35 to 85° by X-ray diffraction (XRD) using a Shimadzu XRD 6000 diffractometer with CuKα radiation operating at 30 kV and 30 mA and with a step scan of 0.02° and a scan speed of 0.2° min$^{-1}$. To collect the patterns, nanoparticles were deposited on the silicon substrate by dripping the alcoholic colloidal dispersion onto the substrate at room temperature and waiting for the solvent to evaporate.

UV–Vis: AgNPs were transferred to a 1.0 cm path length quartz cuvette, and the measurements were made on a Jasco V 660 UV–Vis spectrometer. A typical experiment scanned the wavelength range from 300 to 800 nm. Background adjustments were made using anhydrous ethanol as a blank.

DLS and Zeta Potential: Particle size measurements of the AgNP mixture were conducted with a zetasizer (Nano-ZS90, Malvern, UK) in disposable cuvettes and the average hydrodynamic diameter was determined by taking an arithmetic average of 3 runs. DLS experiments were performed at room temperature and a fixed angle of 173° equipped with a 50 MW 533 nm laser and a digital autocorrelator. In this method, the number-average values obtained were compared to the size distributions of the AgNPs. The surface charge of AgNPs was determined by zeta potential measurements with the same equipment. All experiments were carried out in triplicate and the results presented are the average measurements of the runs with standard deviation.

SEM: SEM images were recorded at 5 kV using a FEG Zeiss Supra 35-VP. The samples were prepared by placing three drops of the diluted colloidal dispersion in anhydrous ethanol onto carbon-coated copper grids (200 mesh, PELCO® Center-Marked Grids, Hawthorne, CA, USA) and dried at room temperature for one day.

HRTEM: The size and morphology of AgNPs synthesized were investigated by the HRTEM TECNAI F 20 Microscope operating at 20 kV. Samples were deposited by placing one drop of dilute colloidal dispersion on a carbon-coated grid and drying at room temperature for one day.

### 2.3. Evaluation of Antimicrobial Activity
2.3.1. Definition of Inhibition Halo

The disk diffusion method was performed in accordance with the *National Committee for Clinical and Laboratory Standards* (Performance standards for antimicrobial disk susceptibility tests; Approved Standard—Eighth Edition. NCCLS 13 document M2-A8, 2003a.) with modifications by Hosida et al., 2018 [45]. The strains of *Candida albicans* (*C. albicans*) (ATCC 10231), *Streptococcus mutans* (*S. mutans*) (*ATCC25175*), *Lactobacillus casei* (*L. casei*) (IAL#523), *Enterecoccus faecalis* (*E. faecalis*) (ATCC51299) and *Actinomyces israelli* (*A. israelli*) (ATCC 12102) were reactivated in SDA agar (Sabouraud Dextrose, Difco, Le Pont de Claix, France) and BHI (Difco) for 48 h at 37 °C in aerobiosis and microaerophilia, respectively, for each microorganism. Then, 5 colonies of each species were placed in BHI broth individually and incubated at 37 °C for 18–24 h. An aliquot of 300 µL of each bacterial suspension (optical

density of 0.6 and absorbance of 550 nm) was homogenized with 15 mL of BHI-agar at 45 °C. After gelling of the culture medium, sterilized paper discs were placed on the surface of the agar medium. A 5 μL aliquot of AgNP solution (Ag:PVP (1:1) and Ag:PVP (4:1)) was placed on the paper discs. For each experiment, a 0.2% chlorhexidine gluconate (CHX) solution was also used as a positive control. The plates were kept for 2 h at room temperature to enable diffusion of the solutions and then incubated at 37 °C for 24 h. Two measurements of each inhibition halo were measured with the aid of a digital caliper and the averages were calculated. The inhibition halo data (agar diffusion test) were heterogeneous and were submitted to Student–Newman–Keuls analysis.

### 2.3.2. Minimum Inhibitory Concentration (MIC)

The microdilution method was performed according to Clinical Laboratory Standards Institute guidelines (CLSI: M27-A2 and M07-A9). AgNP samples were diluted in deionized water in geometric progression, from 2 to 1024 fold. Subsequently, each AgNP was diluted (1:5) in RPMI 1640 medium (Sigma-Aldrich) for *C. albicans* (ATCC 10231), and in Brain Heart Infusion (BHI, Difco, Le Pont de Claix, France) for *S. mutans* (ATCC 25175). The 24 h culture inoculateswase was adjusted to the standard turbidity equivalent of 0.5 McFarland in saline (0.85% NaCl) [46]. The suspensions of each strain were diluted (1:5) in NaCl (0.85%) and further diluted (1:20) in RPMI 1640 for *C. albicans* or BHI broth for *S. mutans*. Each microorganism suspension (100 μL) was added to the wells of microtiter plates containing 100 μL of each nanocomposite concentration. Microtiter plates were incubated at 37 °C, and MICs were determined visually with the lowest concentration of AgNPs without microorganism growth after 48 h [47]. After 48 h, the contents of each well were plated on SDA (for *C. albicans*) or BHI agar (for *S. mutans*) to determine the minimum fungicidal concentration (MFC) and the minimum bactericidal concentration (MBC) of the solutions against the strains tested. Assays were performed in triplicate on three independent occasions.

### 2.3.3. Cytotoxicity of AgNPs

NIH/3T3 fibroblast cells were cultured under standard cell culture conditions in Dulbecco's modified Eagle's medium (DMEM) supplemented with 10% fetal bovine serum (FBS), penicillin, and streptomycin, at 37 °C, 100% humidity, 95% air and 5% $CO_2$. Cells were subsequently seeded into 96-well plates ($10^4$ cells/well) and incubated for 24 h under standard cell culture conditions to enable cells to adhere before adding the solutions. Afterward, several dilutions of the silver solutions in sizes of Ag: PVP (1:1) and Ag:PVP (4:1) were applied to the cells. Cell viability was evaluated by the assay of 3-(4,5-dimethylthiazol-2-yl)-2,5-diphenyltetrazolium bromide (MTT) after 24 and 48 h. For this purpose, the culture medium and the dilution of the solutions were removed from each well and 100 μL of MTT solution (0.5 mg/mL) in DMEM without FBS (1:10) was added to each well. The MTT solution was removed after 4 h of incubation, the formazan crystals were dissolved in 100 μL of isopropyl alcohol. The plate was left at room temperature in a dark chamber for 30 min on a rotary shaker. The absorbance of the plates will be evaluated at 570 nm using an Elisa reader (Expert 96, Asys Hitch, Eugendorf, Austria).

### 2.3.4. Statistical Analysis

Statistical analyses were performed using SigmaPlot (SigmaPlot 12.0, Systat Software Inc., San Jose, CA, USA). Data passed normality (Shapiro Wilk's test) and were submitted to a two-way analysis of variance, followed by Fisher's LSD test, adopting a significance level of 5%.

## 3. Results and Discussions

### 3.1. Mechanism of Formation of AgNPs

The main reducing agents used in the reduction of $Ag^+$ ions in aqueous and non-aqueous solutions are sodium citrate (Turkevich method), sodium borohydride ($NaBH_4$),

NN-dimethylformamide (DMF), hydrazine, glucose, and alcohols [48]. Among others, the most used for the synthesis of AgNPs is the alcohol reduction process, due to greater morphological and size control [49]. In this method, alcohol acts as a solvent and reducing agent, forming AgNPs (Equation (1)) [50]:

$$Ag^+ + \frac{1}{2}CH_3CH_2OH \xrightarrow{\textit{Heating}} Ag^0 + H^+ + \frac{1}{2}CH_3CHO \tag{1}$$

$$n\,Ag^+ + x\text{ reducing agent } \rightarrow \left(Ag^0\right)_n + \text{oxidation products} \tag{2}$$

$$\left(Ag^0\right)_n + Ag^+ \rightarrow \left(Ag^0\right)_{n+1} \tag{3}$$

According to Tatarchuk et al., 2013 [51], the formation of AgNPs occurs through the reduction of $Ag^+$ ions with the transfer of an electron from the reducing agent to the metallic ion (Equations (2) and (3)), which is then followed by nucleation and growth of particles. The step described in Equation (2) occurs with rapid nucleation, in which, in the first minutes of the synthesis process, a percentage of the Ag+ ions is reduced and transformed into nuclei or particles. The step described in Equation (3) consists of the coalescence of particles, followed by their growth. This increase in particle size occurs with a decrease in the reduction process due to a smaller amount of $Ag^+$ ions, suggesting that coalescence or Ostwald maturation is the main growth process of these particles (Figure 1).

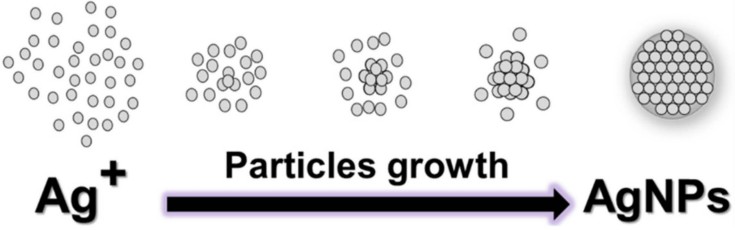

**Figure 1.** Schematic representation of the mechanism of formation of AgNPs.

However, the processes described in Equations (2) and (3) lead to the agglomeration of AgNP clusters that can be avoided with the use of a stabilizing agent, which adsorbs on the surface of the nanoparticles, forming a layer that prevents coalescence [52]. For this purpose, it is possible to use PVP that has nucleophilic atoms (Lewis bases) with high affinity for electrophilic atoms ($Ag^+$ Lewis acids), and sufficiently long organic chains since these properties enable the creation of steric impediment, preventing interactions between the particles [53].

PVP is commonly used as a stabilizing agent that selectively induces the formation of AgNPs and stabilizes the colloidal suspension through steric stabilization [54]. In addition, there is a displacement of the electron pair from the nitrogen atom to the oxygen atom in the carbonyl group (C=O), forming a partial negative charge located on the oxygen atom and a partial positive charge located on the nitrogen atom as shown in Figure 2a.

The reaction mechanism occurs through the coordination of silver ions with the non-bonding electrons of the nitrogen/oxygen atom of the pyrrolidine ring [55,56]. Subsequently, there is a chemical reduction between the metallic ion and the PVP in which the Ag+ ion receives an electron from the carbonyl group and forms AgNPs as shown in Figure 2b [57]. In this method, the $Ag^+$ interacts with PVP, forming Ag (PVP)$^+$.

Due to these interactions, metallic particles will cap upon nucleation and $Ag^+$ ions will stabilize with the complex compound. This stabilization of silver ions reduces the nucleation process and produces larger particles [58]. The key role of the surfactant is to prevent the aggregation of AgNPs. The steric effect arising from the long polyvinyl chain of PVP on the surface of AgNPs may contribute to their anti-agglomeration, as shown in Figure 2c [59].

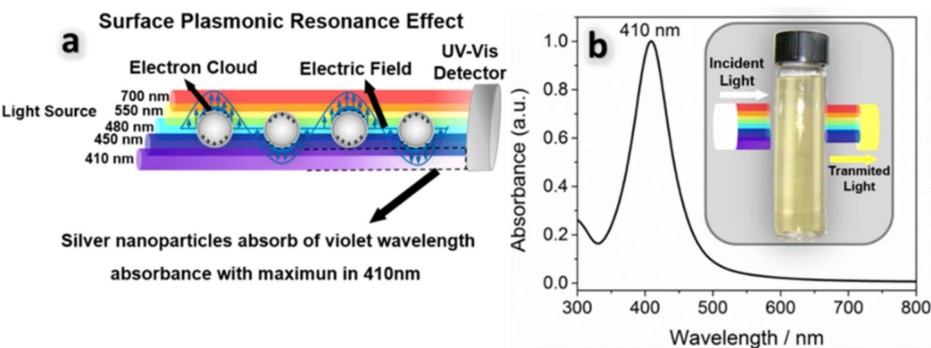

**Figure 2.** Stabilization of AgNPs by PVP. (**a**): Representation of the displacement of the electron pair from the nitrogen atom to the oxygen atom in PVP. (**b**): Mechanism for the formation and protection of AgNPs by PVP. (**c**): AgNPs coated by PVP.

*3.2. Physicochemical Properties of AgNPs*

Qualitative analysis of the presence of AgNPs in the solution can be achieved using UV–Vis analysis. The interaction of particles with electromagnetic radiation provides information such as the stability of colloidal solutions, and structural, morphological, and size distribution parameters.

In this context, white light is composed of various shades of red, orange, yellow, green, blue, indigo, and violet light. Therefore, the colors are the result of the species absorbing selected portions of the visible light spectrum. For example, the solution of AgNPs is yellow and strongly absorbs light at 410–440 nm (purple and blue light) (Figure 3b) [60].

**Figure 3.** Optical result of colloidal solutions of AgNPs (**a**) surface plasmonic resonance effect of AgNPs; (**b**) UV–Vis spectra of AgNPs (at molar ratio of 1:1 Ag:PVP).

Figure 3b shows a symmetric surface plasmonic resonance (SPR) band with maximum absorption at 410 nm characteristic of spherical AgNPs (see Supplementary Material Figure S2). Studies reported by Hah et al., 2003 [38] and Lee & Oh (2014) [44] also demonstrated that the maximum UV absorbance of AgNPs at 410 nm.

This result confirms that the $Ag^+$ ions were reduced to $Ag^0$ by the alcohol reduction process. According to Mie's Theory, only one SPR band is expected in the absorption spectrum of spherical nanoparticles, while two or more SPR bands are expected for anisotropic nanoparticles, depending on the specific shapes of the particles. In addition, the symmetrical shape of the plasmon band can indicate a relatively sharp particle size distribution [61–63].

The solution of the synthesized AgNPs exhibited strong radiation absorption near the wavelengths of 410 to 450 nm due to the s-p (conduction band) and d-s (interband) transitions of electrons, respectively. The s-p transitions depend on the shape and size of the particle. This is a unique property of nanoparticles that is because the s-p (conduction) electrons are largely free to move throughout the nanoparticle, and their energies are therefore sensitive to the shape and size of the box that contains them [64,65].

The excellent light-trapping and local electromagnetic-field-enhancing properties of surface plasmons have been further researched in the field of plasmonics [66], opening a wide range of applications in cancer therapy, photovoltaic devices, catalytic activity, sensors, etc. [67–69].

The structural and morphological results of AgNPs coated with PVP with a stoichiometry of 1:1 Ag:PVP are shown in Figure 4. X-ray diffraction (XRD) in Figure 4a of the sample with a molar ratio of 1:1 Ag:PVP shows peaks at 38.2°, 44.4°, 64.7°, 77.6°, and 81.7°, which can be indexed to the (111), (200), (220), (311), and (222). Different samples showed diffraction peaks related to the face-centered cubic (FCC) structure of metallic Ag (PDF card 04-0783) [70] (see Supplementary Materials Figure S3).

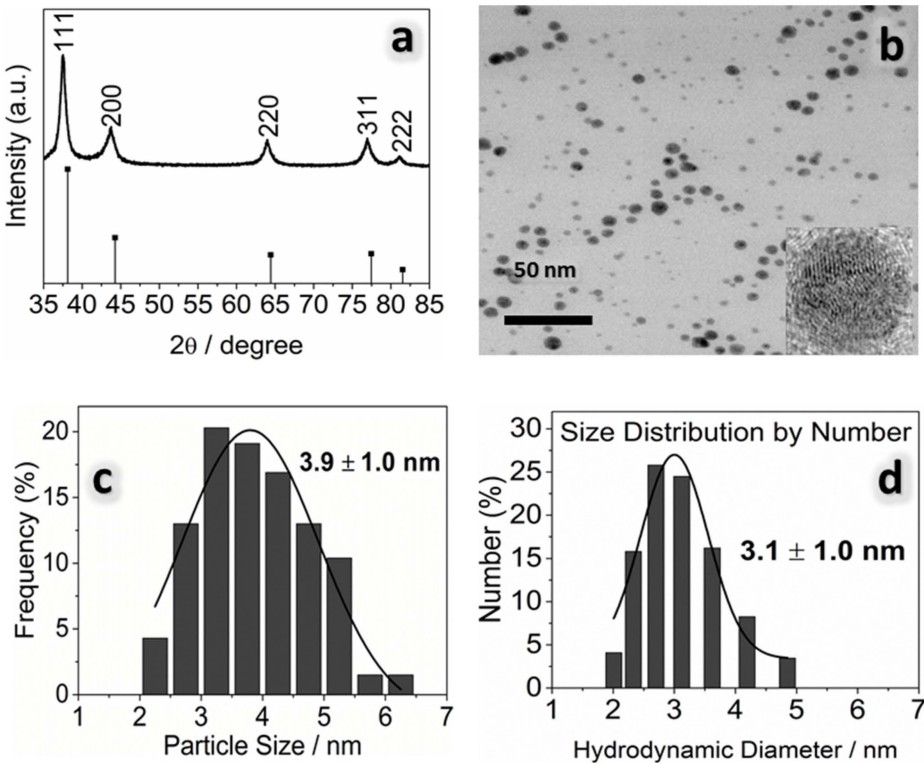

**Figure 4.** Structural and morphological results of colloidal solutions of AgNPs (1:1 Ag:PVP). (**a**) XRD pattern of AgNPs; (**b**) transmission electron microscopy micrographs of AgNPs; (**c**) histogram of transmission electron microscopy of AgNPs; (**d**) size distributions by the number of silver nanoparticles by DLS technique.

The HRTEM analysis of the sample (Figure 4b) showed that the AgNPs have a uniform spherical shape and it is also possible to verify that the PVP forms a layer around the AgNPs, making them more dispersed and preventing them from agglomerating. However, a small number of aggregates is formed during the reduction process, and a smaller amount of polymeric chain interacts with the surface of the formed nanoparticles.

These particles have an average size of 3.9 nm and a narrow particle size distribution (Figure 4c). PDI data confirm the formation of uniform particle sizes when the values were below to 0.7 [71]. The results obtained were expected, since the addition of the PVP stabilizer protected the nanoparticles from possible agglomeration and growth. The methodology used resulted in nanoscale particles, with narrow size distribution and

crystalline phases whose results are in agreement with those described by Lee et al., 2015 [44] (see Supplementary Materials Figures S4 and S5).

Based on the results, it can be stated that not only can the appropriate concentration of PVP prevent AgNP agglomeration due to the amount of reduction of [Ag$^+$] but also particle growth inhibition, once PVP is strongly adsorbed on the nanoparticle surface. This process can be described according to the nucleation and growth theory of nanoparticles. Liu et al., 2020 [72] suggest that the period for seed formation is a factor for nanoparticle size and size distribution. When seeds are formed rapidly over a brief period, the size of the nanoparticle becomes small, and the size distribution becomes narrow. Radicals are species more reactive than aldehydes (Equation (1)) for the reduction process, seeds are formed and a rapid seed formation in ethanol could lead to the formation of small nanoparticles and a narrow size distribution.

In this study, we present a comprehensive comparison of the key characteristics of silver nanoparticles (AgNPs) obtained through various synthesis methods. The table displays information on particle size, observed morphology, and, notably, the concentration of AgNPs. The samples with different proportions of Ag:PVP prepared in our work were compared with published data, as shown in Table 1.

**Table 1.** Comparison of size, morphology, and concentration of AgNPs, and reaction conditions for different methodologies applied to the synthesis of silver nanoparticles in this study with published data.

| Method | Size Particle/nm | Morphology | Solution Concentration | Reference |
|---|---|---|---|---|
| Alcoholic | 3.9 ± 1.0 | Spherical | 45 mM | This study |
| Alcoholic | 651 ± 91 | Spherical | 180 mM | This study |
| Polyol | 48.8 ± 3.3 | Nanocubes | 16 mM | [6] |
| Alcoholic | 4.6 ± 0.8 | Spherical | 40 mM | [73] |
| Alcoholic | 10–15 | Spherical | 1 mM | [66] |
| Turkevich | 30 ± 3.9 | Spherical | 7 mM | [23] |
| Polyol | 15.6 ± 8.30 | Spherical | 6 mM | [17] |
| Turkevich | 21 | Spherical | 4 mM | [42] |
| Green synthesis | 25 ± 2.0 | Spherical | 0.3 mM | [74] |
| Green synthesis | 20 | Spherical | 1 mM | [75] |
| Green synthesis | 27.8 ± 3.13 | Spherical | 1 mM | [76] |
| Turkevich | 40 | Spherical | 0.25 mM | [77] |

Silver nanoparticles have attracted significant interest due to their unique properties and versatile applications across diverse fields. In our research, we have developed a highly promising synthesis method that distinguishes itself from other approaches by producing silver nanoparticles with precise size control and high concentrations.

Attaining silver nanoparticles with a controlled size and a high concentration is paramount for tailoring AgNPs to specific applications. Our proposed synthesis method overcomes challenges faced by traditional approaches, ensuring uniformity in particle size and facilitating the generation of concentrated AgNP suspensions. This enhanced control over size and concentration enables us to fully exploit the potential of silver nanoparticles in various practical applications. Our findings contribute to the advancement of nanoparticle synthesis and pave the way for impactful applications in diverse fields.

Complementary analyses to determine the size were performed using DLS analysis and the results obtained showed particles with a size of 3.1 ± 1.0 nm (Figure 4d) similar to HRTEM analysis (see Supplementary Materials Figures S6–S8). The explanation for this result is related to the fact that a lower Ag:PVP stoichiometry forms a thicker layer on the surface of metallic nanoparticles, preventing aggregation and resulting in smaller particles [78].

To evaluate the effect of different MW (10 K, 40 K and 360 K) and molar concentrations (Ag:PVP) (1:4, 1:2, 1:1, 2:1 and 4:1) on the average size values of AgNPs, the DLS technique used is shown in Figure 5. The results show that the MW did not influence the size of

particles obtained in the same Ag:PVP stoichiometric condition. Chou et al., 2004 [79] obtained results that also showed no influence on of AgNP particle size when different PVP MW were used. Du et al., 2008 [80] explained that due to the newly formed AgNP presented for the same PVP mass per unit area covering their surface, there are the same interactions between Ag⁺ ions and the carbon atoms oxygen/nitrogen, avoiding greater nucleation and growth. Thus, polymeric chain size does not influence the particle size if the stoichiometric condition is maintained [81,82].

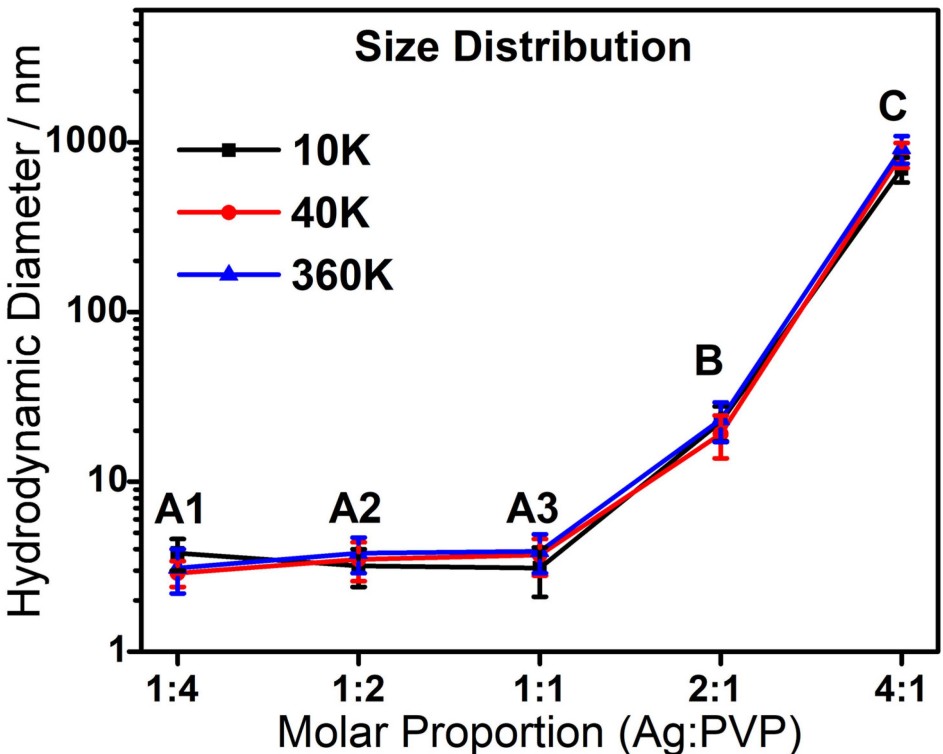

**Figure 5.** Average hydrodynamic diameters of AgNPs obtained by the DLS technique with different average molecular weights of PVP and in different conditions of Ag:PVP molar ratio.

The results of different molar concentrations show that there is an influence on the size of particles formed. Figure 5 shows that the increase in silver concentration in solution (1:4, 1:2, 1:1, 2:1, and 4:1 Ag:PVP), resulting in an increase in the particle size of AgNPs. Therefore, with this methodology, it was possible to control particle sizes and generate particles with an average particle size of $3.20 \pm 0.41$, $3.50 \pm 0.30$, $3.40 \pm 0.40$, $24.8 \pm 3.27$, and $822 \pm 113$ nm, respectively. This result shows that there is no linear growth and a 4:1 Ag:PVP composition had a larger average particle size compared to other compositions as well as a wide particle distribution.

Figure 5 shows the tested range of silver concentrations in the solution, as indicated in "A1–A3" (1:4, 1:2, 1:1 Ag:PVP), did not show any statistically significant differences in the size of silver nanoparticles. This suggests that within this molar ratio range, the monomer of polymer PVP (polyvinylpyrrolidone) did not significantly influence the size of the silver nanoparticles.

However, when we extended our investigation to the "A3-B-C" solution range (1:1, 2:1, and 4:1 Ag:PVP), a noticeable trend emerged. As the Ag:PVP ratio increased, the size of the silver particles also increased. This indicates that a higher concentration of silver ions in relation to the monomer of PVP led to the formation of larger silver particles. The observed increase in particle size can be attributed to the availability of excess silver ions, promoting their aggregation and subsequent growth. These findings underscore the importance of the Ag:PVP ratio in controlling the size of the produced silver nanoparticles, offering insights

into the optimal conditions for nanoparticle synthesis and their potential applications in various fields.

In the 4:1 Ag:PVP group, where there is a smaller amount of PVP, the AgNO$_3$ precursor dissolves faster in ethanol. When the reduction process is fast, PVP molecules do not have sufficient time to coat these newly formed AgNPs, thus preventing agglomeration and resulting in large particle sizes. This can be explained by nucleation and growth mechanisms based on La Mer's model. Nishimoto et al., 2018 [83] reported a schematic representation of the nucleation and growth mechanisms at various PVP concentrations. Nucleation only occurs when the concentration of Ag exceeds the critical supersaturation level. When the concentration of the monomers falls below the critical supersaturation level, only particle growth occurs. With a higher concentration of silver particles in the Ag:PVP (4:1) sample, there is more interaction between silver ions and the PVP polymer molecule, and the surface of the nanoparticles was not covered by a capping agent. Therefore, nanoparticle aggregation and uncontrolled particle growth occurred. As a result, irregularly shaped and large size particles were obtained.

### 3.3. Antimicrobial Properties of AgNPs

To perform the microbiological and cytotoxic tests, two samples were chosen (1:1 and 4:1 Ag:PVP and MW 10 K) because they were statistically different in size ($p < 0.05$). These samples were Ag:PVP (1:1) with a concentration of 7.6 g L$^{-1}$ and Ag:PVP (4:1) with a con-centration of 30.4 g L$^{-1}$.

The antimicrobial effect of AgNPs was evaluated using the minimum inhibitory concentration test (MIC), in which the lowest concentration of a chemical responsible for limiting the visible growth of a bacterium, that is, which has bacteriostatic activity, is verified. Concomitantly, the agar diffusion test was performed, in which a microorganism is challenged against a biologically active substance in a solid culture medium and relates the size of the challenged microorganism's growth inhibition zone to the concentration of the tested substance.

Figure 6 shows the average results of the inhibition zone formed by the microorganisms Enterecoccus faecalis (*E. faecalis*), Lactobacillus casei (*L. casei*), *Streptococcus mutans* (*S. mutans*), *Candida albicans* (*C. albicans*) and *Actinomyces israelii* (*A israelli*) formed by the Ag:PVP (1:1) and Ag:PVP (4:1).

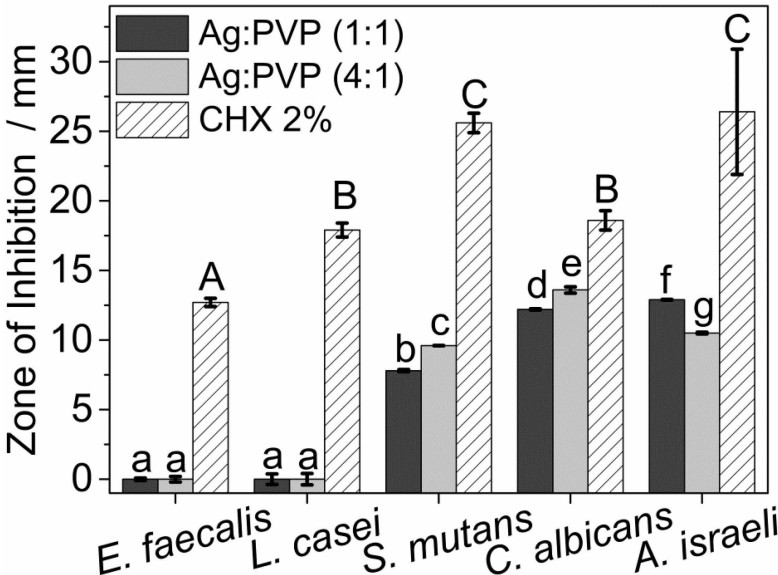

**Figure 6.** Means and standard deviation (bars) of the diameters of the step halos for different microorganisms. Different letters show difference statistics between groups.

The results showed the antimicrobial effect of AgNPs against different microorganisms (*S. mutans*, *C. albicans* and *A. israelii*) since inhibition halos were observed for these microorganisms and the antimicrobial potential of the nanoparticles is proportional to the size of the halo of inhibition formed [84].

The choice of microorganisms occurred because *Enterococcus faecalis* is a persistent organism that, despite constituting a small proportion of the flora in untreated root canals, plays an important role in the etiology of persistent periradicular lesions after endodontic treatment [85]. *Lactobacilli* sp. represent a characteristic group of oral bacteria that numerically comprise a minor component of the oral microbiota. Despite these low numbers of lactobacilli in the oral cavity, initial findings show that lactobacilli may be correlated with caries formation [86,87]. *Streptococcus mutans* is one of the many etiological factors of dental caries, as it is a microorganism capable of acquiring new properties that enable the expression of pathogenicity determinants, determining its virulence in specific environmental conditions [88]. *Candida albicans*, which causes candidiasis, is a common infection of the skin, oral cavity and esophagus, gastrointestinal tract, vagina and vascular system of humans; and, lastly, *Actinomyces* sp. has been frequently cultured from the root canals of teeth with primary and post-apical periodontitis treatment [89,90]. These bacteria are thought to be associated with persistent extraradicular infections, lack of periradicular healing, and cases of failed endodontic therapy [91–93].

AgNPs have well-established antimicrobial activity against Gram-positive/negative microorganisms, fungi, protozoa, and some antibiotic-resistant viruses and strains [94]. However, Gram-negative microorganisms have a greater antimicrobial effect compared to Gram-positive microorganisms [73]. This can be explained by the difference in cell wall thickness between Gram-positive (30 nm) and Gram-negative (3–4 nm) microorganisms, which are mainly composed of peptidoglycan [95,96]. Moura et al., 2012 [97] showed in their study that there was a greater halo of inhibition for the microorganism *S. aureus* compared to *E. coli* and they associated this result with the load of AgNPs.

However, the specific response of each microorganism depends on its metabolic activities. As the cell membrane surface of microorganisms is positively charged, there is an electrostatic attraction between AgNPs that has a negatively charged surface (Ag:PVP (1:1): $\zeta = -36.3 \pm 3.80$ mV; Ag:PVP (1:1): $\zeta = -32.4 \pm 2.60$ mV), thus facilitating the penetration and diffusion of AgNPs into the microorganisms.

The antimicrobial activity of AgNPs is well established; however, its mechanism of action is not yet fully understood. AgNPs bind and adhere to the cell membrane causing direct damage [98], or, upon permeating the cell membrane, they dissolve and release silver ions. Then, once inside the microorganism's cell, AgNPs leads to DNA damage, mainly through the production of ROS [99], and can also affect the electrochemical gradient of protons through the respiratory process [100], interrupting ATP synthesis, leading to the death of the microorganism cell [101].

The antimicrobial activity of the AgNP solutions was evaluated by estimating the minimum inhibitory concentration (MIC) and the minimum fungicidal concentration (MFC). The results showed that AgNPs have fungicidal properties against the microorganisms tested at low concentrations (Table 2).

**Table 2.** Results of MIC and MFC/MBC of silver nanoparticles with different sizes.

| Species | Ag:PVP (1:1) | | Ag:PVP (4:1) | |
| --- | --- | --- | --- | --- |
| | MIC (mg mL$^{-1}$) | MFC/MBC (mg mL$^{-1}$) | MIC (mg mL$^{-1}$) | MFC/MBC (mg mL$^{-1}$) |
| *C. albicans* | 9.400 | 300.9 | 9.400 | 9.400 |
| *S. mutans* | 601.9 | 300.9 | 75.23 | 75.23 |

The greater susceptibility of planktonic cells of *C. albicans* compared to planktonic cells of *S. mutans* can be explained mainly by structural differences in their cell membranes [102]. The membranes of bacteria are negatively charged due to anionic phospholipids, while

the charge on the membranes of fungi is neutral. As the AgNPs synthesized in the present study have a negative surface charge, this may explain the decreased susceptibility of *S. mutans* and the need for a greater concentration to inhibit its growth [103].

Similar results were obtained by Monteiro et al., 2011 [104] and Wady et al., 2012 [105] in which the authors observed that AgNPs exhibited fungicidal activity against *C. albicans* and that AgNPs cause depolarization and fungal membrane rupture with an increase in the release of carbohydrates such as glucose and trehalose into the intracellular environment (compounds that protect proteins and biological membranes from inactivation by stress conditions), leading to damage to the fungal cell structure and inhibition of the budding process of the microorganism [106].

According to Abbaszadegan et al. 2015 [107], the antimicrobial action of AgNPs depends on the external surface charge of the particles influenced by the stabilizers and agents used in the synthesis. In the MIC assays (Table 1), both strains of each species were susceptible to AgNPs. AgNPs promoted a 100% reduction in planktonic growth of microorganisms in concentrations ranging from 9.40 to 602 mg mL$^{-1}$.

These differences in the composition and especially in the cell wall thickness of each microorganism may have contributed to the increase in MIC values and values observed for *S. mutans* planktonic cells in relation to *C. albicans* cells.

AgNPs also have an effect against *S. mutans* bacteria and similar results were found by Martínez-Robles et al., 2016 [108]. The bactericidal activity can be attributed to the high surface areas in the nanoparticles and the ability of AgNPs to bind to membranes [109,110]. This action occurs through electrostatic interactions and through interactions with the bacterial cell membrane adhesion system, which allows strong links between the AgNPs with the amino, hydroxyl, and thiol groups present in the cell membrane, which leads to the death of the microorganism.

Therefore, to try to explain the difference between microbial activities, it is necessary to evaluate the cell walls of different microorganisms, since the cell walls of the microorganisms are the first point of contact between nanoparticles and microorganisms [111]. The cell wall of *C. albicans* is composed of 1,3-β-glucan, 1,6-β-glucan, chitin, and mannoproteins [112] and has a thickness of 25–75 nm [113], while *S. mutans* has a cell wall composed of polysaccharides (rhamnose and glucose) [114] and a thickness of 300–400 nm [115].

The in vitro cytotoxicity assay is the test to assess the biocompatibility of any material for use in biomedical devices. Cytotoxicity tests consist of placing the material directly or indirectly in contact with cell culture and verifying cell alterations through different mechanisms. Cell viability was determined by the MTT assay, which is one of the most widely used tools in cell biology to measure cell metabolism [116]. The test is based on the reduction of MTT, a yellow water-soluble salt, by the effect of cellular metabolic activity linked to NADH and NADPH, forming insoluble formazan crystals, blue or purple in color. Blue or purple staining is therefore a quantifier of cell viability [116].

Cell viability data are shown in Figure 7. In this study, greater cell viability was observed for the smaller particle size Ag:PVP (1:1) compared to the larger particle size Ag:PVP (4:1). A decrease in cell viability was identified for both AgNPs at 1, $^{1}/_{2}$, and $^{1}/_{4}$ dilutions at 24 and 48 h, with no statistical difference between them. The other dilutions (1/64 and 1/128) showed greater cell viability for Ag:PVP (1:1) when compared to the other dilutions, but with no significant difference between them ($p < 0.05$), regardless of the period evaluated.

The development of nanoparticles in the technology industry is stimulated due to their innovative properties; however, people are still concerned about the toxicity of these materials [117]. In Figure 7, it is possible to observe that Ag:PVP (1:1) caused a cytotoxic effect with a significant reduction in the number of viable cells in the dilutions with a higher concentration of silver present in relation to the other dilutions. On the other hand, Ag:PVP (1:1) induced significant cytotoxicity from lower concentrations of these AgNPs. The PVP polymer can bind to the surface of silver nanoparticles, resulting in a flocculation process, due to the nitrogen atom present in its molecule [118]. Even with the flocculation process,

the polymer molecule chain keeps the silver nanoparticles separate from each other, and the particles can interact with cells due to their high surface energy and mobility [118].

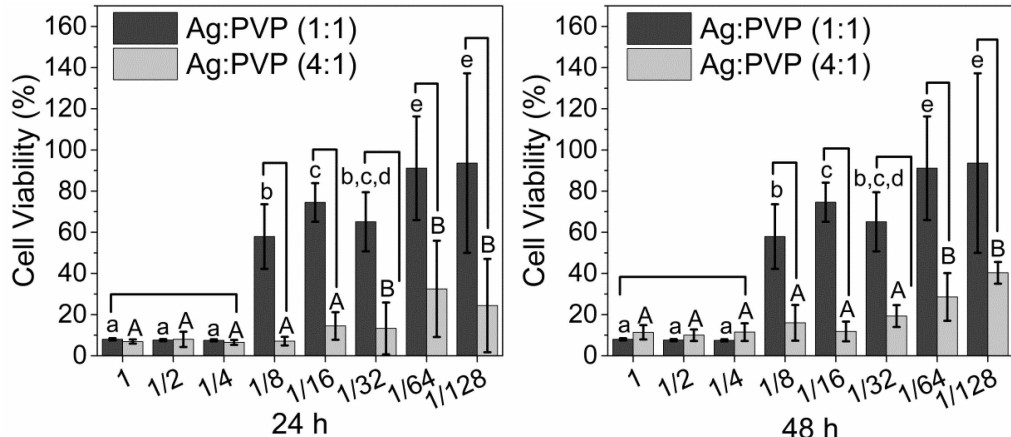

**Figure 7.** Viability of NIH/3T3 fibroblasts determined by an MTT assay at 24 and 48 h. Uppercase and lowercase letters denote the comparison between the AgNPs in each %. Different symbols denote significant differences in AgNPs and time in each %.

El Badawy et al., 2011 [119] report in their study that the toxicity of silver nanoparticles can be partially attributed to the presence of impurities, such as silver ions, reducing agents, and residual stabilizers from the synthesis of silver nanoparticles, because the removal of residual impurities had a strong impact on reducing the toxicity of silver nanoparticles. On the other hand, the presence of the PVP polymer, which forms a core–shell structure with silver nanoparticles [120,121], may have interfered with the interaction process of AgNPs with cells, thus producing the lowest cytotoxic effect when in lower concentrations for Ag:PVP (1:1) compared to Ag:PVP (4:1) due to the size of the silver particle that may have contributed to these results.

In view of the results obtained, new tests should be carried out, given the importance of evaluating the oxidizing activity of these new particles. Among future tests, it would be interesting to evaluate the action of these nanoparticles on oxidative stress [122], biological markers and others.

## 4. Conclusions

Our proposed synthesis method represents a significant advancement in the production of silver nanoparticles (AgNPs), setting itself apart from other methods presented in this study. The presence of polyvinylpyrrolidone (PVP) proved to be crucial in achieving controlled size and high concentrations of AgNPs.

Through our investigations, we demonstrated that PVP acts as a cap, effectively binding with silver ions and nanoparticles, and playing a pivotal role in the synthesis process. The PVP coating effectively prevents agglomeration and disorderly growth of Ag atoms, resulting in the formation of AgNPs with well-defined sizes and shapes.

We found that by adjusting the Ag/PVP molar ratio, we could fine-tune the characteristics of the AgNPs. Increasing the PVP/Ag molar ratio led to enhanced coverage of Ag atoms, yielding AgNPs with smaller sizes. Conversely, reducing the PVP/Ag ratio resulted in fewer Ag atoms being capped, leading to the synthesis of larger-sized AgNPs.

The key advantage of our method lies in the ability to prepare AgNPs with precise size control while achieving high particle concentrations. This level of control over size and concentration is crucial for tailoring AgNPs to specific applications, including their potential use as antimicrobial agents against different microorganisms.

In summary, our innovative synthesis method, leveraging the essential role of PVP, provides a pathway to produce AgNPs with a controlled size and shape, and high concentrations. These tailored nanoparticles have demonstrated promising biological activities

against various microorganisms, opening up exciting opportunities for their application in diverse fields, such as medicine, catalysis, and environmental remediation.

**Supplementary Materials:** The following supporting information can be downloaded at: https://www.mdpi.com/article/10.3390/colloids7040066/s1, Figure S1: Schematic representation of the synthesis of silver nanoparticles in the alcoholic medium; Figure S2: UV–Vis spectra of synthesized AgNPs at different concentrations and average molecular weights of PVP. A: 10 K; B: 40 K; C 360 K; Figure S3: X-ray diffractograms of AgNPs at different concentrations and average molecular weights of PVP 10 K; Figure S4: SEM micrographs of AgNPs and particle size distribution (2:1 Ag:PVP 10 K); Figure S5: SEM micrographs of AgNPs and particle size distribution (4:1 Ag:PVP 10 K); Figure S6: Size distribution by number (%) of the hydrodynamic diameter at different concentrations and average molecular weights of PVP 10 K; Figure S7: Size distribution by the number (%) of the hydrodynamic diameter at different concentrations and the average molecular weights of PVP 40 K. Figure S8: Size distribution by number (%) of the hydrodynamic diameter at different concentrations and the average molecular weights of PVP 360 K.

**Author Contributions:** F.N.S.N. was responsible for conceptualization, methodology, formal analysis, investigation, resources, and writing—original draft preparation. L.A.M. and T.Y.H. were responsible for conceptualization, antimicrobial tests, formal analysis, investigation, resources, and writing—original draft preparation. L.S.R. was responsible for formal analysis (application of statistics) and writing—review and editing. L.F.G., T.J.M. and P.F. were responsible for visualization and writing—review and editing. D.B.B., E.R.C. and A.C.B.D. were responsible for project administration, supervision, funding acquisition, and writing—review and editing. All authors have read and agreed to the published version of the manuscript.

**Funding:** This research was funded by the Brazilian agency Sao Paulo Research Foundation (FAPESP) (grant numbers 2016/17577-7, 2018/16041-1, 2017/17993-3, 2023/11303-6).

**Data Availability Statement:** Not applicable.

**Acknowledgments:** The authors acknowledge the Brazilian agencies Sao Paulo Research Foundation (FAPESP) (grant numbers 2012/07067-0, 2013/23572-0, 2016/019405, and 2023/11303-6) for financial and publication support, and the concession of a scholarship. Special thanks to Research, Innovation and Dissemination Centers (CEPID) (2013/07296-2) for technical support and Nation Council for Scientific and Technological Development (CNPq) (grant numbers 435975/2018-8, and 421648/2018-0). and Coordination of Higher Education Personnel (CAPES—Brazil)—Finance Code 001.

**Conflicts of Interest:** There are no conflict to declare, and if accepted, the article will not be published elsewhere in the same form, in any language, without the written consent of the publisher.

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
