# Peer review of "Facile Synthesis of PVP-Coated Silver Nanoparticles and Evaluation of Their Physicochemical, Antimicrobial and Toxic Activity"

_colloids, doi:10.3390/colloids7040066_

Round 1

Reviewer 1 Report

Comments and Suggestions for Authors

Reviewer comments:

This manuscript reported PVP Coated Silver Nanoparticles were used for biological and toxic activity. However, this manuscript needs major revisions and the following comments will help to increase the quality of the manuscript before publication.

·         Abstract should be more specific based on the findings with recommendation.

·         In the "Introduction" section, general description on the importance of manuscript topic is poor. Therefore, the importance of this work cannot be well recognized from general readers. In order to fix this problem, addition of description on recent development in the field of research topic with citing recent comprehensive papers would be important.

·         Author should include the FTIR information to confirm the functional group in PVP /AgNps.

·         How is the work different or better than those reported earlier? Author needs to highlight this in the introduction and discussion part.

·         Author carefully check the scientific terminology in entire text.

·         Why you choose particular bacteria for antimicrobial activity. Why author not checked fungal pathogens?

·         Why author not checked Antioxidant assay? Author must include the reason in the text with appropriate reference citation https://doi.org/10.1016/j.jksus.2022.102029;

·         Author should be include pathogens virulence factors https://doi.org/10.1155/2022/7234586

·         Extensive editing of English language and style required

·         Need to check consistency in reference the in text 

·         Author should acknowledge the recommendation of present study in the conclusion section.

Author Response

We sincerely appreciate the valuable feedback provided by Reviewer #1. Their comments have been carefully considered, and we agree that major revisions are necessary to enhance the quality of our manuscript before publication.

We appreciate the understanding of Reviewer #1 regarding the constraints we face and their recognition of the dedication of our research team. We firmly believe that our work holds potential to contribute to the scientific community, and we are determined to refine it to meet the highest standards possible.

We would like to express our gratitude to Reviewer #1 for recognizing the dedication of our research team despite the challenges we encounter. We believe that our work has the potential to contribute to the scientific community, and we are committed to refining it to meet the highest standards.

Reviewer 2 Report

Comments and Suggestions for Authors

Dear Authors,

The manuscript has a portion of scientific significance; however, I feel the lack of novelty in this paper. The manuscript needs improvement, as some parts are unclear and hard to read. Some recommendations and questions are below.

Comments:

1.     Could the authors write the added value of this article compared to the literature?

2.     Authors should think about a better structural presentation of results. This part is hard to read. The Table showing the different nanoparticle composition, size distribution, zeta potential etc. would summarize the nanoparticle properties.

3.     The authors stated (line 85) the synthesis of “stable NPs in high concentration” in the introduction as the motivation of this work. How long is the nanoparticle solution stable in time? What is the yield of nanoparticles synthesized with this method?

4.     The authors worked with “diluted samples of NPs” in antibacterial and cytotoxic experiments. What is the concentration or mass of nanoparticles in the stock solution? This is important information for easier comparison of results with the literature.

5.     Could authors specify the letters A, B, C and a, b, c….in graphs? Does it have statistical meaning?

6.     Authors use words such as bacteriostatic and bactericidal. Is the nanoparticle effect on cells bactericidal or bacteriostatic?  

7.     Line 375: probably the wrong Ag/PVP ratio.

8.     Figure S4: Here, the authors show SEM images of nanoparticle size instead of HRTEM ones. Focused on the low resolution of S4 image, could authors describe the image analysis procedure evaluating the nanoparticle size?

9.     The manuscript needs grammar polishing.

Author Response

We sincerely appreciate the thorough revision of the manuscript by Reviewer 2 and the valuable suggestions and comments provided. We acknowledge that it is important to highlight the added value of our article compared to the existing literature.

In response to the reviewer's recommendation, we have made significant revisions to the manuscript. We have included additional comparisons of our data with previously published papers, which help to demonstrate the unique contributions and advancements offered by our study. These comparisons provide a clear understanding of how our findings extend the current knowledge in the field.

Furthermore, we have emphasized the distinctive aspects of our research methodology, experimental design, or data analysis techniques that set our study apart from previous works. This highlights the novel and original aspects of our research, further enhancing its value in the scientific community.

Round 2

Reviewer 1 Report

Comments and Suggestions for Authors

The authors responded to all my comments and the editor can accept the revised version. Thank you

Reviewer 2 Report

Comments and Suggestions for Authors

Dear Authors,

The article was revised satisfactorily by you to improve the quality of the article. All comments and recommendations were successfully answered and integrated into the manuscript.